# Enabling In-Situ Urbanization through Digitalization

**Le Li** [1] **and Tao Song** [2,3,*]

1 Rural Development Institute, Chinese Acadmy of Social Sciences, Beijing 100732, China; nfs-lile@cass.org.cn
2 Institute of Geographic Sciences and Natural Resources Research, Chinese Academy of Sciences, Beijing 100101, China
3 University of Chinese Academy of Sciences, Beijing 100049, China
* Correspondence: songtao@igsnrr.ac.cn

**Abstract:** The bourgeoning of e-commerce in the context of the information era has accelerated the urbanization trend by broaching a new horizon of economic and industrial boosters for rural places, epitomized by a great number of "Taobao Villages" in China. This paper has two objectives: (1) explore the process and mechanism of digitalization enabling rural in-situ urbanization represented by e-commerce; (2) nuance the specific case evidence of Daiji Town, where digitalization enabled in-situ urbanization recently. We build up a theoretical framework for digitalization-enabled in-situ urbanization from the juxtaposition of four interlinked elements: industry, talent, rural governance, and land use. It then analyzed the details and evidence of digitalization enabling rural in-situ urbanization through the case study of Daiji Town. The main conclusions of this paper are as follows: First, digitalization plugs rural areas into production and consumption networks in wider contexts, promoting the transformation and prosperity of rural economies. Secondly, the reverse migration of young generations to rural areas becomes the key to rural in-situ urbanization. Thirdly, digitization materializes the urbanization of rural spaces. Finally, digitalization enables the rural transformation and improvement of urban-rural relations in the Global South, which needs to be further explored.

**Keywords:** digitalization; rural in-situ urbanization; Taobao village; China

## 1. Introduction

The continuous prevalence of e-commerce in the UK, China, Africa, the BRICS countries, and other developed and developing countries is of great significance for the revitalization of rural areas in the world [1–3]. In the past decade, the digitalization represented by e-commerce has been reshaping China's rural society at an unimaginable speed [4]. Our attention to Cao County in Shandong Province stems from a news article vignette in May 2021, in which an online blogger believes that his hometown, Cao County, is more attractive than such modern metropolises as Beijing and Shanghai, attracting more than 260 million video views and over 500 million topics read on the microblog platform about Cao County [5]. The flow dividend has attracted widespread social attention to this little-known small county in the southwestern Shandong Province, China. There are nearly 60,000 online shops in the whole county, driving 350,000 people to start businesses and engage in relevant work. With 151 Taobao villages and 17 Taobao towns in the county and an annual sale of more than 20 billion yuan, it becomes the second largest rural e-commerce cluster in China after Yiwu in Zhejiang Province [6]. Taobao, owned by the electronic retailer Alibaba Group, is the largest e-commerce platform in China, along with eBay and Amazon. "Taobao Village" is a cluster of rural electronic retailers in an administrative village. With more than 10% of households operating online Taobao stores, the total turnover there exceeds 10 million yuan [7]. And when a town has three Taobao villages, or its annual e-commerce sales from them exceed 30 million yuan, and the number of active online shops exceeds 300, it could be termed a Taobao town [8]. In 2021, there will be a total of

7023 Taobao villages and 2171 Taobao towns in China, widely distributed in 27 provincial administrative units [9].

The wonders generated by the combination of e-commerce and China's rural and small towns, with Taobao Village and Taobao Town as carriers, have quickly attracted the interest of the academic community. The existing literature has studied the industrial and production structure [4], employment opportunities [10], community organization and governance [11], lifestyle [12], rural environment [13], and spatial structure [14–16] of Taobao Villages. A vast number of rural geographical scholars have noticed that digital transformation is radically changing the constellations and processes of production, marketing, and consumption in the rural production system [17]. On the one hand, digital information technology has changed the information asymmetry between supply and demand in rural areas and the outside world, making farmers' production and management targeted [18]. On the other hand, it promotes the links between rural industries by reorganizing the original production factors and rural communities [19]. E-commerce has become a technical catalyst for changes in rural industrial structure, employment patterns, and household economies [10]. Moreover, noticeable evidence proves that rurality and everyday mundane life in rural China have changed in response to skyrocketing digitalization [12], especially in burgeoning Taobao villages in China's coastal areas, which serves as a prime example of the global "digital turn". In essence, this change can be seen as the impact of digitalization on the spatial reconstruction of rural economic geography.

The process of drawing urban cores and even distant hinterlands into orbit, thus giving them more of the physical, economic, social, and cultural characteristics usually associated with cities, is one of the most commonly studied processes within scholarship on planetary urbanization in the global South [20–24]. How migration contributes to the urbanization process has attracted more attention among a vast array of theoretical urbanization inquiries [25–27]. Migrants have traditionally been viewed as being responsible for excessive urban growth and urban surplus labor [28–33]. However, since the 1980s, China has been witnessing the emergence of rapid rural industrialization in the form of township and village enterprisesenterprises (TVEs), which not only led to the rapid development of towns but also the huge expansion of villages. Scholars have termed this phenomenon "in situ urbanization" or "urbanization from below", distinguishing it from city-centered urbanization [34–37]. Thus, in-situ urbanization is a process by which peasants harvest non-agricultural employment and citizenization based on the original village and small towns [38] by engaging in rurally diversified economies [39]. This rural economic boom engenders sweeping improvements in physical infrastructure such as roads, electricity, and information technology, as well as transforming the countryside into a "functional extension of the city" [40]. Later on, as economic reforms shifted the focus from rural to urban areas, especially after the 1993 recentralization of public finances, TVEs and townships were phased out of the central stage of the policy agenda in China [41]. Beyond these vignettes of rural urbanization through industrialization, the stories of digitalized in-situ urbanization remain poorly discussed.

Up to date, under the guidance of China's national scheme towards rural revitalization, a new round of rural urbanization is fundamentally reconstructing rural settlements [42,43]. The above research showed that e-commerce, by using the time-space compression effect, had built a bridge between rural producers and urban consumers and accelerated the reallocation of capital, personnel, wealth, and information between urban and rural areas [44]. Rural production and lifestyle are "embedded" into urban life through the Internet, forming a benign interaction between rural and urban areas. This interaction is a complex and comprehensive process that reconstructs the rural economic structure, social form, governance mode, and spatial structure [45,46], showing the typical significance of rural urbanization.

It is noteworthy that there are already some discussions about the rural in-situ urbanization in Taobao Village. Luo and He believed that Taobao Village is the product of e-commerce in the information age, revealing a new type of rural in-situ urbanization [47]. This process reconstructs the social, economic, and physical spaces of rural areas. Lou and

Hu proposed the concept of "urbanization in real time", trying to explain that urbanization can be realized without completely changing the form of rural settlements through real-time communication technology and rapid transportation [48]. Lin constructed an analytical framework of e-urbanism, which consists of three interwoven levels: the ICT infrastructure and production network, the social network and power relationship, and the urban form and land use [49]. Although the above research has noted the signs of urbanization in Taobao Village, it still lacks empirical and qualitative evidence of the impacts of digitalization on urbanization. Much as digitalization has a central place in geographical inquiries, a holistic and nuanced analysis of digitalization-enabled rural in-situ urbanization coupled with erasing the binary between urbanization and rurality is urgently needed.

Digitalization is looming large as a leviathan booster of global rural restructuring by enabling urbanization. Nonetheless, there is still a lack of a framework based on a comprehensive perspective to describe and depict the complex and overall process of the role mechanism of digitalization enabling in-situ urbanization, especially in the context of China's national scheme towards rural revitalization. This paper has two objectives: (1) explore the process and mechanism of digitalization enabling in-situ urbanization represented by e-commerce and try to verify them through microscopic empirical research. (2) to nuance the specific case evidence of Daiji Town, where digitalization-enabled urbanization is starting recently. It is structured as follows: After the introduction, we will propose a theoretical framework for digitalization-enabled urbanization in the second part. The third part will introduce methods and data collection. The profile of our case evidence will be shown in the fourth part. It then elaborates on the nuanced process and mechanism of digitalization-enabled urbanization in Daiji Town. We conclude with some key highlights and policy discussions.

## 2. Digitalization-Enabled Urbanization in Rural China towards a Conceptual Framework

In this subsection, we curate the digitalization-enabled urbanization framework from the juxtaposition of four interlinked elements (Figure 1), which echo recent scholarly buzzwords and concomitantly construct the digitalization-enabled urbanization theoretical landscape.

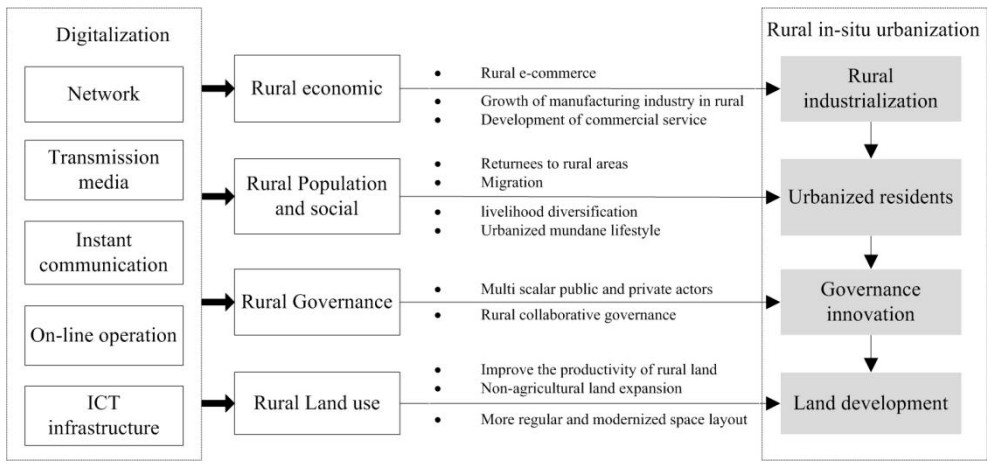

**Figure 1.** Digitalization-enabled rural in-situ urbanization.

### 2.1. Digitalization-Enabled Industrialization

Digitalization is regarded as broaching a new horizon of economic and industrial boosters for rural places. Evidence proves that the digital transformation is radically changing constellations and processes of production, marketing, and consumption in the agrifood system, helping farmers deliver safe, sustainable, and quality food [50]. However, it can also be more adaptive to climate change [51]. In addition to upgrading traditional agriculture, digitalization-enabled rural growth is more prominent and empowered by

the so-called "rural industry", which refers to the dominant non-agricultural industry, either in manufacture or services, in one township and surrounding villages [13]. As China's evidence shows, family-based workshops facilitated early rural industrialization, which then turned into urban agglomerations [52]. To date, the overall development of rural industry in rural China has been rapid and spatially uneven, with highly developed industries in the eastern region and less developed industries in the middle and western regions [53]. The Taobao village, featured by diffusing e-commerce platforms in rural China, is regarded as a prominent and effective means of revitalizing rural areas and narrowing the rural-urban gap by both academia and the government [13], through transforming the structure, processes, and strategies of most industries but also spawning entirely new enterprises and industrial chains. In addition to China's case evidence, Germany's tales also proved that digital platform-oriented firms are relatively often located in urban regions, whereas digital manufacturers are of significance for rural industrialization [54].

### 2.2. Digitalization-Enabled Urbanized Residents

Instead of passive donation recipients and price takers, farmers serve as active entrepreneurs who take risks and innovate to adapt to the transition from farmers to urbanized residents [55]. In this vein, rural residents are enabled through diverse innovative practices and knowledge-intensive skills [56]. The diversification of the economic base of rural communities has often been read as a shift toward post-productivism [57,58]. Accordingly, farmers, immigrants, and second-home owners living in rural areas reap momentum equipped with digital skills and sufficient funding in this mixed post-productivism milieu and thus play a proactive agency role in the rural progression featured by the intertwined and mutually constitutive production of hybridity, diversity, and heterogeneity [59]. Indeed, the "silent revolution" of digitalization in rural China is transforming and reconstituting both rural economies and civil society. As Wang et al. illustrated, the multifaceted nature of rural transformation wreaked by e-commerce disintermediation includes livelihood diversification, the de-linking of livelihoods from land, the de-localization of livelihoods, cultural and social changes, and the disintermediation of e-commerce, all of which amount to complicating the deepening of the rural-urban divide. In sum, digitalization-enabled urbanization is a flexible relational process that is embedded in the mundane everyday practices of smallholder farmers, indigenous rural dwellers, and e-commerce employers and employees [60].

### 2.3. Digitalization-Enabled Governance Innovation

In the multiple digitalization-related definitions, it is common to refer to the application of digitalization to innovate governance through various technologies by multi-scalar actors. The technology industry's ascent as a force advancing urban governance ideology is particularly important because it permeates cities' political administrative structures and reframes public discourse about urban issues in technology-centered terms [61,62]. For example, the "Smart Territory" is termed the one that seeks to solve public problems through solutions based on technology in the framework of a partnership between multiple participants from different sectors, both public and private [63,64]. Accordingly, the digitalization dimension of a policy is not linked to technological determinism but instead refers to a broader context of abilities to ramp up governance innovation at some specific sites by both multi-scalar public and private actors. Moreover, governments at multiple scales around the world have diversified their policy portfolios to galvanize multiple economic, social, and environmental governance targets. Liang and Li noted that the Chinese government has issued a series of governance innovations and policies to promote the development of the digital economy [65]. Jamil unpacked digitization policy-related challenges in Pakistan, including a lack of policy evaluation and refinement, a lack of focused research, and inappropriate allocations of funds at federal, national, and sectoral levels that affect wide-ranging digitalization in Pakistan [66]. In this paper, we examine how



digitalization has enabled governance agencies in rural areas through governance service delivery, community engagement, and multi-scalar civic participation.

*2.4. Land Reconfigurations Driven by Digitalization*

"Rural-urban digital divide" has been coined since the 21st century to picture and characterize spatial inequalities during the digital era [67]. Rosen and León developed the "digital growth machine" rationale to reveal the rural-urban land transition in pursuit of capturing land-related profit through traditional land expansions and digitalization intensification strategies [61]. Beyond simplistically replicating traditional land growth in a new digital sector, the digital growth machine regards the land as a panoply of physical and digitized manifestations that range from goods and services to data flows, whereby land use forms and intensification have been thoroughly reconfigured. To curb such spatial inequalities, which are of significance for rural land development, digital technologies and applications are widely discussed and aimed at improving the productivity of rural land, advancing rural economies, and favoring the inclusion of rural communities in cultural, social, and political activities. Digitalization through such digital technologies as sensors, cloud computing, smart grids, social media platforms, etc. has been actively engaging in the digitalized circularity of urban metabolism [68], cohesion policies, and sustainable land management [64].

Drawing on this conceptual assemblage, this approach delivers the opportunity to transcend the unitary dichotomy between rural/urban areas by integrating all potent elements into a holistic framework of digitalization-enabled urbanization. In tracing the subfield of this dynamic through the following case evidence, the paper places an emphasis on how various elements constitute and sustain the in-situ urbanization driven by digitalization.

## 3. Data and Method

Data supporting this research is collected and assembled in three distinct but interlinked ways: (1) first-hand data, which is mainly collected from in-depth interviews with local government officials, village cadres, online shop operators, general staff of e-commerce companies, and local villagers. (2) the second-hand data, including relevant government documents and plans, publications published in journals, newspapers, and the Internet, and media interviews and reports. (3) In addition to the spatial data, we also visually demonstrate the spatial layout of Taobao Village in Daiji Town by deploying the Arcgis model to discern land use changes caused by the development of the e-commerce industry there.

The authors of the manuscript conducted ethnographic fieldwork in Cao County and Daiji Town mainly between 2022 and 2023, with some short follow-up trips in 2023, in order to make an in-depth, conceptual, and theoretical description and analysis of the internal operation of rural socioeconomic structure and the process of digitalization that enabled in-situ urbanization. Semi-structured and open-ended interviews with more than 50 government officials, village cadres, enterprise managers, ordinary residents, factory workers, and villages were completed accordingly. Snowball sampling was conducted in these ways: first, through local connections by the first author, who lived in Daiji Town for about 2 months and worked in an embroidery studio for 3 weeks; and second, via official connections the other coauthor built up through the role of consultant for a project conducted in Shandong Province. Table 1 lists the key respondents' information in this study. From the interviews with government departments of Cao County and government officials of Daiji Town, we mainly learned information from three aspects: (1) the role of government departments in the development of digitalization and the e-commerce industry in Daiji Town; (2) the process of development, operation, and management in the E-commerce Industrial Zone of Daiji Town; and (3) the governments' planning and toolkit for the future development of e-commerce in Daiji Town. Moreover, we conducted interviews with relevant village cadres of Dinglou and Sunzhuang Village, which are

the earliest to become Taobao villages in Daiji Town, trying to figure out the following three issues: (1) how e-commerce originated and rapidly developed in villages; (2) how the development of e-commerce changed and reshaped rural landscapes; and (3) how digitalization impacted rural governance. The study protocol was approved by the local government of Daiji Town. The study also recruited more than 20 village participants through face-to-face intercept interviews. All of these villagers were engaged in digitalized e-commerce businesses as bosses or employees. Several related questions were asked: (1) the basic situation and the production and management links of online shops; (2) the impacts of e-commerce and digitalization on their livelihood, income level, and living habits; and (3) their feelings about the surrounding environment and visions for the future of the village. 4. Profile of Daiji Town in Shandong Province.

**Table 1.** Key actors interviewed by the study.

| Category | Name | Organization/Identity | Code |
|---|---|---|---|
| Government officials | Mr. Zhang | Cao County Government, E-commerce Service Center | GO1 |
| | Mr. Xie | Cao County Government, Bureau of Industry and Information Technology | GO2 |
| | Mr. Li | Daiji Town Government | GO3 |
| | Mr. Li | Daiji Town Government | GO4 |
| Village cadres | Mr. Sun | Sunzhuang Village | VC1 |
| | Mr. Sun | Sunzhuang Village | VC2 |
| | Mr. Reng | Dinglou Village | VC3 |
| Entrepreneurs of leading enterprises | Mr. Hu | Executive Director, Chenfei Clothing Co., Ltd. | EL1 |
| | Mrs. Meng | Design Director, Chen Fei Clothing Co., Ltd. | EL2 |
| | Mrs. Zhou | Manager, Qingsheng Performance Clothing Co., Ltd. | EL3 |
| Villager | Mrs. Cui | Owner of Hanfu online shop | V1 |
| | Mr. Zhao | Owner of performance costumes online shop | V2 |
| | Mr. Sun | Owner of performance costumes online shop | V3 |
| | Mrs. Liu | Staff of fabric store | V4 |
| | Mr. Zhang | Staff of Yunda Express | V5 |

Our case Daiji Town is located in the southeast of Cao County, Shandong Province, with a total area of 45 square kilometers and a total population of about 47,000 (Figure 2). Aquaculture, planting, and processing of agricultural and sideline products were the leading industries in the 1980s. In the late 1990s, Cao County actively opened up a number of textile factories and established vocational and technical schools to train textile skills and promote economic development; therefore, the villagers of Daiji mastered the skills of making performance costumes. In 2009, China's "online shopping" craze was in a period of rapid development, and the increase of cultural performance activities created a huge segment of the market for performance costumes. Daiji Town became one of the earliest Taobao Towns in China, and all 32 administrative villages under its jurisdiction have been identified as "Taobao Villages". According to data from the Ali Research Institute, a small town such as Daiji Town in Cao County, where every village has been rated as the "Taobao Village" for four consecutive years, is unique in the whole country [9].

As the development models of Taobao towns are so diverse that it seems impossible to use just one or two of them to fully summarize the more than 2000 Taobao towns in China, our case represents a typical example of digitalization-enabled urbanization in rural areas, which should be applicable to many small towns in China for the following reasons:

Firstly, e-commerce in Daiji Town is booming. Dominated by the performance costumes and Hanfu manufacture, the e-commerce market there is getting rapid growth, currently with 18,000 registered online shops, and about 70% of performance costumes and 30% of Hanfu in China's e-commerce market are produced in Daiji Town. In addition, e-commerce sales here have also shown explosive growth in recent years. According to interviews with the local government staff, the e-commerce transactions in Daiji Town reached

9 billion yuan in 2021, with about 80% of the local residents engaged in e-commerce-related jobs (GO1).

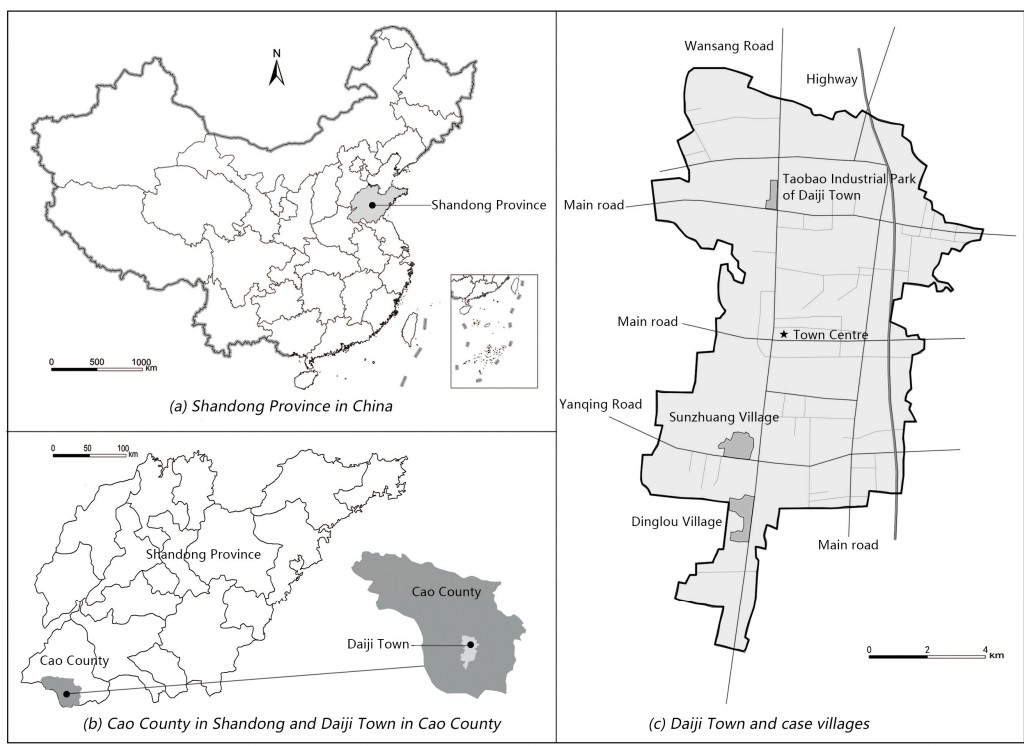

**Figure 2.** The location of the case study area.

Secondly, the initial development of e-commerce in Daiji Town is completely a result of spontaneous market rationality, which is dominated by local entrepreneurs to a large extent. And most Taobao villages and towns in China follow a similar bottom-up development path.

Thirdly, previous studies have shown that residents in villages close to urban centers can be more successful in diversifying their livelihoods by combining the non-agricultural employment opportunities nearby [69]. In our case, Daiji Town is a traditional agricultural town far away from the city center. With no national or provincial highways but only one county road passing through the border, this town does not have any advantage in its traffic conditions. And most of the villages in China have a similar development background and foundation. Therefore, our case study can help us understand how digitization empowers the urbanization of these agricultural towns far from the periphery of big cities.

## 4. Digitalization-Enabled Urbanization in Daiji Town

### 4.1. Digitalization, E-Commerce, and Industrialization

The traditional offline transaction is conducted through face-to-face communication: the buyers go to a specific place, communicate with the sellers, and complete the exchange of information, goods, and funds through direct contact. In contrast, e-commerce delivers a digital internet platform through which buyers browse the details of goods, communicate with sellers, and complete the payment. The sellers deliver the goods to the buyers through the logistics system shortly after confirming the payment. In short, e-commerce enables buyers and sellers to exchange information and trade over an infinitely wide geographical area, breaking the dependence of trade on spatial location and distance [70].

Our case evidence will demonstrate the nexus of digitalization, e-commerce, and industrialization by witnessing the vicissitudes of the performance costume market, which is a "niche market" with a small audience. Before the development of e-commerce, the sales of costumes in rural areas depended to a large extent on the market in the county,

making the trade basically limited to the local area. As a result of the narrow sales channels, the costume manufacturing enterprises in Daiji Town were generally small in scale and low in output. Then the emergence of digitalized e-commerce broke the information asymmetry between small merchants and the outside market, allowing producers to directly have access to the super-sized potential market with the possibility of sufficient market demand to support all types of products. Through e-commerce, Daiji Town was connected to national markets, and buyers from different regions started to buy their products, making the sales increase significantly. Shortly afterward, thanks to the "ripple effect" in China's rural society [71], the information and experience of successful online trades spread rapidly among relatives and neighbors, and more and more rural households joined the online business.

"Twenty years ago, we usually took our costume samples to the wedding photography shops in the county town to sell them door-to-door. In case of product overstock, we never produced too much at one time. And in order to sell the costumes, the farthest place I went was Heilongjiang Province. But so hard as I tried, the number of costumes sold out in a year could be about 100 at most (EL3)".

"In 2009, one of my friends came to visit me and mentioned "Taobao", a platform where people could sell costumes without renting shops or going out for promotion. My wife decided to open an online shop there immediately. When the Children's Day on June 1st was coming, the first customer took the initiative to contact us online and ordered over 200 sets of student uniforms at one time. As well as excited, we were worried about having such a big order that we could not finish the costumes manufacturing in time, so we invited two neighbors to join us and in this way, we finished the first electronic order in Daiji Town (VC3)".

Digitalization creates a data-driven production mode, which changes the traditional way of production organization in rural areas. The development of social media and Web 2.0 provides great support for the transformation of e-commerce from product-oriented to social and customer-centric [4]. Digital communication has enabled designers, consumers, and small businesses to have direct contact opportunities with each other. The information and data fed back by the market became an important basis for the production of enterprises. Once market demand changes, the production cooperation between small and medium-sized enterprises and family workshops could be quickly adjusted and reorganized into a new production network. This model was beneficial for economic organizations with limited production capacity and marketing capacity in rural areas to avoid potential risks. What is more, the information exchange has also inspired designers and small enterprises to constantly create new styles and new categories of products to obtain new markets.

"I joined dozens of online communities established by the Hanfu enthusiasts, including WeChat group, QQ group and Douban group, to learn about the clothing style and price preference from young people. We would publish our design sketches in the communities to get real-time feedback from potential consumers, including the clothing styles, colors, fabrics and price. And the Hanfu enthusiasts could exchange any of their ideas with me here. Moreover, we would launch online polls and the most popular and potentially best-selling products would be massed-produced in our factory(V1)".

Whereas the structure, processes, and strategies of most industries have been reshaped by e-commerce, thousands of entirely new enterprises and industrial chains have sprung up in this trend. With the increasing number of online businesses in Daiji Town, many rural households began to set up enterprises of raw material and accessory businesses for performance costumes and Hanfu, enabling the clothing enterprises and family workshops to have local access to all the raw materials needed for production (Figure 3). The raw materials for these enterprises mainly came from Zhejiang Province, a major textile production area in China. At the same time, enterprises with some new functions also emerged. By providing professional webcasting services, online marketing strategies, and online or offline employee training for clothing manufacturing enterprises, they played key roles in accurately coordinating relationships between the upstream and the downstream,

producers and consumers (Figure 4). The development of Taobao villages has enabled a large number of small, scattered businesses in rural areas of China to be connected and clustered into modern urban business assemblages through ICT and logistics networks [4].

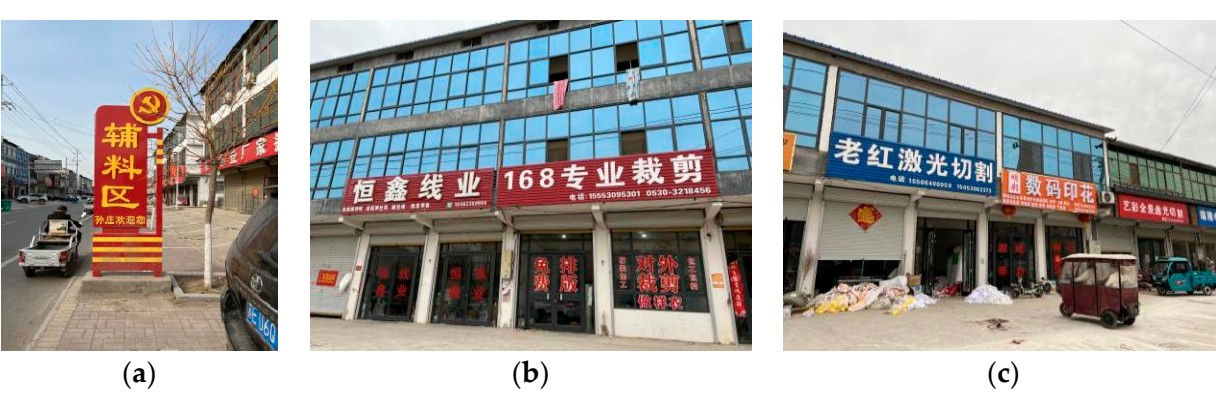

| (a) | (b) | (c) |

**Figure 3.** Raw material and accessory businesses for performance costumes and Hanfu (Source: photos taken during the fieldwork). (**a**) raw material and accessory district, (**b**) Thread and tailor's shops, (**c**) Embroidering and textile printing shops.

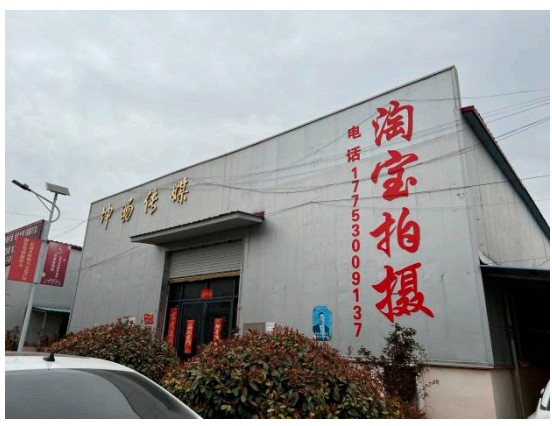

**Figure 4.** A media company provides photo services for online shops (Source: photos taken during the fieldwork).

"On both sides of the Wansang Road, there are as many as 60 to 70 companies selling cloth and more than 200 companies running subsidiary material businesses. Moreover, here also distribute companies mainly engaged in designing, pattern making, embroidering, and textile printing. The Sunzhuang Village plans to expand the original scale and improve the upstream industrial chain of performance costume production. In the future, with the sales channels of performance costumes and Hanfu broadened, the demand and scope of the market for subsidiary materials will be larger, which urges us to speed up the pace of opening new cloth stores. Our goal is to become the largest subsidiary material market of costumes in the north of the Yangtze River (VC2)".

*4.2. Reverse Migration and Social Changes Associated with Digitalization*

The unprecedented worldwide digitization has not only unleashed economic miracles for a vast number of villages but also begat a series of profound changes to the social structure of these rural areas. In the context of the declining economic status of rural areas caused by the binary of urban-rural and the constant outflow of young and middle-aged population into urbanized areas, attracting talents back to the countryside has always been an important scheme of rural revitalization in China [72]. The booming development of digitalized e-commerce has spawned a larger number of jobs with considerable incomes.

As the neo-classical economic theory implies, higher incomes and better employment opportunities will drive population migration [4,73]. As a result, a reverse migration of the population began to take place, i.e., a large number of young people who had moved from rural areas to urban areas returned to the countryside [74].

According to our interview, Daiji Town used to be a big labor export town, with about 60% of the labor force migrating to cities to seek a better life. Since the rapid development of e-commerce in the 1990s, lots of new employment opportunities have been created, attracting the younger generation in rural areas back to their hometowns. Some of those jobs are directly related to e-commerce, such as those of online shopkeepers, website designers, Taobao photographers, and so on. while the others are indirectly related, including both the upstream and the downstream, such as logistics and distribution. According to local government officials, by the year 2021, there would have been more than 7000 migrant farmers and over 700 college graduates in Daiji Town coming back to start businesses, with 2 PhD students and 14 Master students included, which is very rare in rural areas.

"I used to work in clothing factories run by Koreans in Qingdao and Yantai with a salary of more than 3000 yuan a month. Now I and my wife operate a store on Taobao, which enables us to earn 150,000 yuan a year without too hard work. We can make no less money through business on Taobao than working outside and we can take better care of the elderly and children at home(V2)".

A large number of young people returning to the countryside have reinvigorated the languishing rural areas and boosted the local economy. Most of them coming back to start their own businesses are equipped with the necessary information technology as well as many other urban skills, such as business consciousness, which enable them to re-feed the local farmers in the fields of culture, information, and technology to a certain extent [75]. They played an important catalytic role in the formation of Taobao villages.

A very typical case is Dr. Hu and his wife, Mrs. Meng. As the first group of young people in Daiji Town to return to their hometown to start businesses, they won the title of "Shandong e-commerce entrepreneurial leader." Their enterprise has grown into a well-known local leading enterprise in Hanfu, which integrates the industrial ecosystem of the design department, standardized production workshop, finished product display, original new product release, traditional e-commerce, and new media live e-commerce, driving the employment of more than 200 local villagers (EL1).

"In October 2014, when my husband and I went back to our hometown to visit relatives, we saw many people in the village engaged in e-commerce. Soon after, we decided to set up our own store on Taobao. In 2017, the "Hanfu craze" began to rise, which We thought was a good momentum of cultural consumption. In the second year, we quickly turned to the "battlefield" of Hanfu". From imitation to independent design, we set up several Hanfu stores on major e-commerce platforms and even cross-border e-commerce platforms (EL2)".

"I majored in fine arts as an undergraduate, so I feel very happy to engage in clothing design since I can not only make use of the professional knowledge I had learned, but also develop my own interest at the same time. And the other knowledge learned in the university is also a helper for me to do this, equipped with which I can learn things faster than other villagers, such as the layout design and the operation and maintenance of platform (EL2)".

Young entrepreneurs coming back to their hometowns call on more young people to start businesses in rural areas through their neighborhoods, peer groups, and other interpersonal networks. The reverse migration of the population drives the inflow of capital and information technology, which is a phenomenon of social revival promoted by the development of e-commerce. We saw this word in a vignette from the local government to young migrant workers:

"Your hometown has embracing you with ushering in young entrepreneurship boom. Running around outside is not good as running stores on Taobao, which is a sunrise

industry with a prosperous future. Let's keep up with the pace of time, sit at home, tap the keyboard and play the movement of youth entrepreneurship (V3)".

There is a clear dual epistemology between urban and rural areas, in which the rural corresponds to a natural way of life while the urban relates to a realized center [76]. Digitalization has intensified the inextricable connection between urban and rural areas, which has enabled the modern lifestyle of the city to extend into the countryside. With the change in working patterns resulting from the transition from the traditional production mode, the villagers' concept of time also took a new turn. As office workers who have fixed working hours, most consumers buy goods during off-work hours or at night. In this circumstance, villagers who used to "work at sunrise and rest at sunset" are forced to change their timetable. In order to provide real-time online consulting services, they often have to work until midnight, making the distinction between family, leisure, and work blurred. As described by (V5) and observed by us, the landscape of urban life quietly emerges in Daiji Town:

"Around 4:00 p.m. is the time for the daily centralized delivery of online merchants and at that time, the crossroads in the center of the town are always jammed with cars, express trucks and fabric transport vehicles, which is exactly like the same "evening peak" in the big city (V5)".

"Restaurants springs up like bamboo shoots in the town. On summer nights there will be 20 or 30 barbecue stalls and midnight snack stalls, making the streets still brightly lit at 11 or 12:00 at night. It is hard to imagine that before the development of e-commerce, there were only 2 or 3 restaurants in Daiji Town and the streets were almost empty after 8:00 p.m. (GO4)".

*4.3. Collaborative Governance*

After being embedded in rural society, digitalization brought about the differentiation of internal interests in rural areas through far-reaching and extensive commercial processes, promoting the deconstruction and reconstruction of rural governance structures. In the context of the unique institutional conditions in rural China, special attention is paid to the interests and power relations of village organizations, villagers, and floating populations (including floating elites), as well as local governments in rural production and reproduction spaces [77]. Rural stakeholders build trust through dialogues, further, form a consensus commitment to the direction of rural development, and organize joint actions on the basis of existing institutional arrangements, village resources, and knowledge [78].

In the case of Daiji Town, the market increment dominated by performance costumes was still in its infancy. Being not sensitive to this kind of local change, all kinds of stakeholders did not have a definite risk perception or accurate market expectations. However, the extremely low entry threshold enabled the participants in the rural e-commerce industry to enter continuously and tentatively. And when e-commerce became the main source of income for rural households in Daiji Town, which coincided with the goal that the rural collective and government had been committed to promoting farmers to get rid of poverty and become rich, the governments began to realize the opportunities and benefits of rural e-commerce development. Therefore, building Taobao villages has been supported and promoted by the local governments as a way of rural revitalization. Eventually, the local governments, along with the rural collectives and industry associations, will play active roles in the decision-making process of rural governance, guided by the consensus to promote the development of e-commerce.

The local governments have the power to allocate resources and the organizational ability to mobilize urban and rural development, and they are the main institutional designers and key providers of starting conditions for rural governance. The government of Daiji Town has set up a Taobao Development Office, which is especially responsible for affairs related to the development of rural e-commerce (GO2). And they have made "Preferential Policies for Encouraging the Development of E-Commerce," which mainly include simplifying administrative procedures for registering e-commerce companies, appropriately

reducing or exempting taxes, and providing financial incentives for e-commerce enterprises. They have also given large-scale financial investment to the infrastructure construction of Daiji Town, making its level higher than that of ordinary rural areas. Specifically, to meet the demand of tens of thousands of online operators for instantaneous electric power and network speed during the peak season, the governments have carried out power grid upgrades and fiber-to-the-home projects, which have improved the carrying capacity of rural information infrastructure. Moreover, to meet the logistics needs, they have also carried out a comprehensive transformation of the main roads in this town, including road resurfacing and widening (GO3).

"At first, we were very afraid of power failure and network outage. In this case, customers couldn't find and contact us when they need to. The conditions in our town were poor and sometimes the electric power would be cut off when it rained heavily. The network was also unstable, so our home not only had access to the Unicom Network, but also the Mobile Network and Telecom Network in case of the sudden network outage(V3)".

The governance affairs of rural collectives have changed from the traditional organization of irrigation, self-defense, dispute mediation, mutual assistance, entertainment, and clan activities to accelerate the development of e-commerce. The village collectives have assumed the dual identities of "decision-makers of the family" and "agents". Under the background of collective ownership of rural land in China, the collectively operated construction land is owned by rural collectives and can be used for rural production and business activities. In our case, the collective operational construction land in Sunzhuang Village, Dinglou Village, and other villages was jointly developed, providing sufficient space for the construction of public service agencies for e-commerce and a number of express logistics companies. In addition, the village collectives cooperated with the deployment of the government as organizers to carry out specific tasks to promote rural development, including organizing villagers to join in the construction of infrastructure such as roads and communications and conducting e-commerce skills training for villagers (VC1).

Driven by economic interests, the villagers' subjectivity awakened rapidly, and they began to actively participate in village affairs instead of being indifferent as in the past. Through their participation in voting, villagers elected the elites of e-commerce to take important positions in rural organizations and enter the core of rural governance as representatives of their own interests.

"In the transition election of the Village Party Branch Committee and Villagers' Committee of Dinglou Village in Daiji Town in October 2014, the villagers actively participated in the democratic vote. Finally, Ren Qingsheng, who was believed to do well in e-commerce and have the ability to lead the villagers to become rich through e-commerce, was elected as the village party secretary. And before engaged in e-commerce, he was just an electrician in the village and worked outside for a long time (GO3)".

With the surge of e-commerce merchants, bad competitive behaviors among enterprises began to appear in the market, such as underselling and placing orders maliciously to give moderate or bad reviews. The market began to exhibit bad competitive behavior among enterprises, such as selling at low prices, maliciously issuing orders to medium and poor ratings, and so on. In addition, due to poor tax supervision of rural e-commerce, many merchants are evading taxes. More professional organizations or groups were urgently needed to join the governance network. In this circumstance, an e-commerce industry association was established in Daiji Town in 2016, with a local e-commerce elite as its president and the performance costume processing enterprises and merchants on Taobao in this town as its members voluntarily. The e-commerce association strengthened coordination, cooperation, and exchange among e-commerce enterprises, shared industry market information, strengthened industry self-discipline, and enhanced the ability of enterprises to resist risks. Despite the many merits that e-commerce development brings, one caveat should be warranted: tax evasion is the dark side of e-commerce, and our case in Daiji Town could never eschew this challenge. Thus, the association can do little about fixing the tax evasion, nor can the local government. The e-commerce association standardized

the market behavior of enterprises. Firstly, we will ensure the quality of products in Daiji Town by organizing technical seminars for different enterprises. Secondly, we will conduct propaganda and education for the e-commerce merchants and maintain market order through the communication of our association in order to avoid mutual price depression and ensure the benign development of the market (GO3).

*4.4. Land-Use and Spatial Changes*

The flows of capital, personnel, goods, and information affect the function and form of rural landscapes [79]. With the blooming development of rural tourism, rural e-commerce, and other industries, many villages are becoming multi-functional and post-industrial [80]. The rural space is constantly reconstructed in the process of creative destruction, which is characterized by functional mixing and land cover change. The case of Daiji Town shows that urban spatial organization patterns and land use layouts appear in rural areas and form rural urbanization landscapes because the rural non-agricultural production space has an increasingly important position and the rural low-density land use landscape also changed accordingly.

Under the current rural land regime in China, the land directly related to villagers includes homesteads and contracted land [81]. The land use nature of the contracted land is generally cultivated land, whose main function is crop planting, and it cannot be changed at will as it is protected by the basic farmland system. And the use and transformation of homesteads are more flexible; they can not only be used for self-built housing but also be applied for rental with the consent of the villagers. This provides clues for understanding the evolution logic of non-agricultural production space in Daiji Town, from family workshops and factory sheds along the street to modern industrial parks.

According to our interviews with villagers, in the initial stage of the development of e-commerce in Daiji Town, the family workshop was the general production mode for households in this town. Villagers used their own houses or built simple factory sheds in the homestead courtyard as the main production space for clothing production and sales (Figure 5). The family had formed an intergenerational division of labor production pattern composed of youth, middle age, and old age, among which the young couples were mainly responsible for clothing design, manufacture, and online shop operation, while middle-aged parents participated in the relatively simple links in the process of clothing manufacture and clothing packaging. As all the production links were completed in the villagers' own houses, a kind of production space roughly mixed with residential function was formed and scattered in the countryside, leaving great public security risks, especially fire.

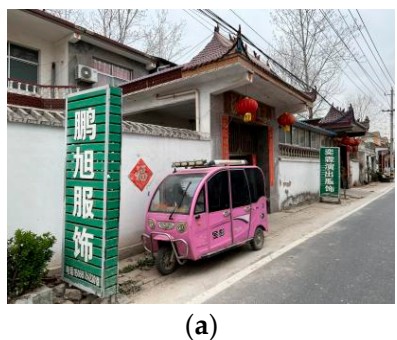
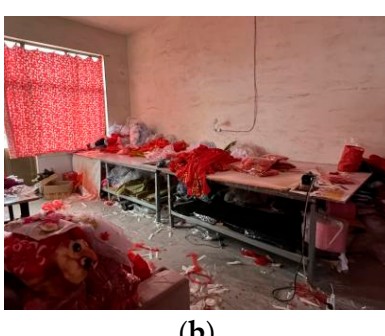
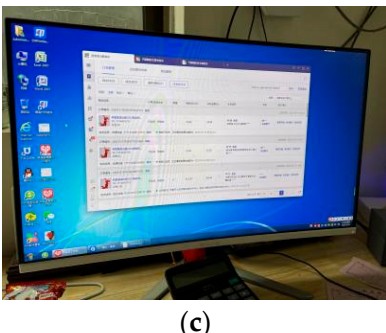

(**a**)　　　　　　　　　　　　　　(**b**)　　　　　　　　　　　　　　(**c**)

**Figure 5.** The family workshop that mixed residential and productive functions (Source: photos taken during the fieldwork). (**a**) Villager houses, (**b**) Production space, (**c**) computer runs online shops.

In order to adapt to new forms of production and consumption, complex spatial transformations often take place in rural areas. The Dinglou and Sunzhuang Villages, located in the south of Daiji Town, serve as an excellent epitome of the spatial transformation of a traditional village into a Taobao village (Figure 6). The villagers and rural collectives

have explored a spatial development model of "joint construction," in which 22 factory sheds have been built spontaneously on both sides of the main roads in the village, known as "Taobao Street". The funds and land for the street construction came from villagers, and the organization of the construction and allocation of the right to use production space were in the charge of rural collectives. The part along the street of these factory sheds is shops for goods display and offices for e-commerce customer service, and the rear is space for production and storage. A large number of e-commerce practitioners are gathered on both sides of the "Taobao Street", i.e., the Wansang Road, making various productive factors such as information, capital, technology, and so on interact rapidly here.

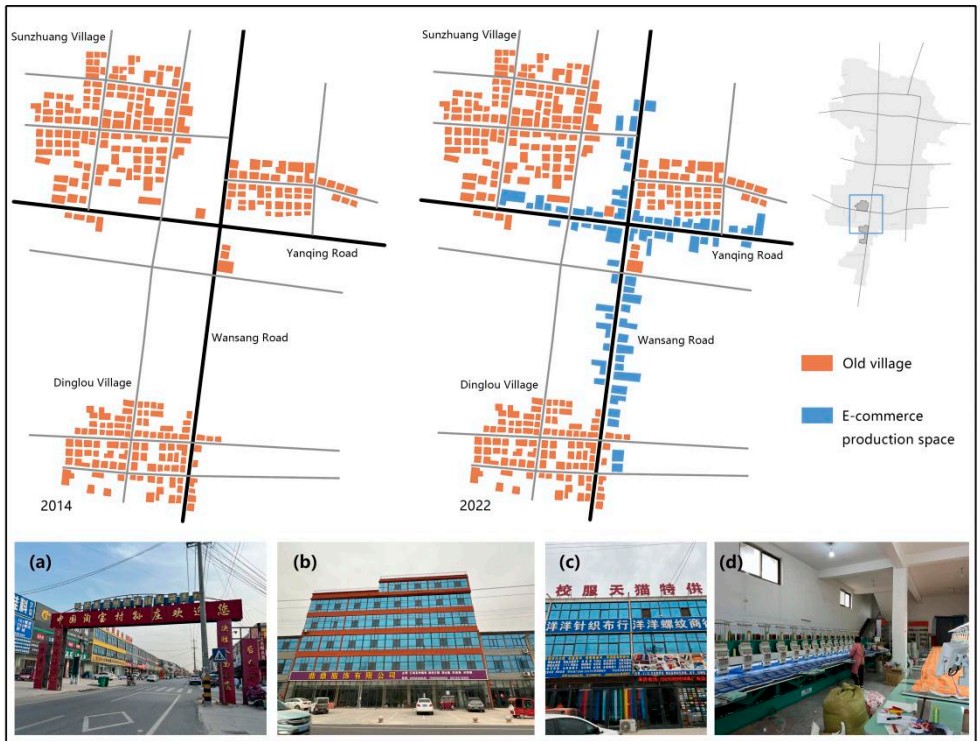

**Figure 6.** Spatial transformation of Dinglou and Sunzhuang village. (**a**) Wansang Road: known as the "Taobao Street"; (**b**,**c**): shops along the main street; (**d**): a small garment processing workshop for online shops (Source: author's mapping and photography).

However, the Taobao clusters built by villagers and rural collectives are proven to be informal. The property of informality is the result of rational economic strategy and "a spontaneous and creative response to the inability of the state to meet the basic needs of the poor" [82]. The land for the factory sheds comes from the contracted land of villagers, so the original agricultural production space is replaced by non-agricultural production space, which seems illegal in the context of China's national scheme to strictly protect the farmland. In addition, informality has also led to a series of safety and environmental issues in rural areas, challenging the government's governance capability.

With the continuous expansion of the e-commerce industry in Daiji town, the informal spatial expansion mode is proven to be unsustainable due to limited yearly land quotas and more strictly up-down environmental monitoring by China's central government. In 2017, the government allocated more than 300 mu of state-owned construction land for the construction of the Taobao industrial park in Daiji Town (Figure 7). According to our fieldwork, the industrial park was constructed in two phases, including the rural e-commerce integrated service building, 48 standardized factories of 1000 square meters, 220 shops, 18 logistics centers, and staff quarters and canteens. The supporting facilities and producer services of the industrial park were significantly attractive to new entrepreneurs and entrepreneurs who pursue high quality. It can be found that e-commerce has affected

the spatial layout of Daiji town to a certain extent. With a more standardized and modern new layout, it can adapt to the new forms of production and consumption better.

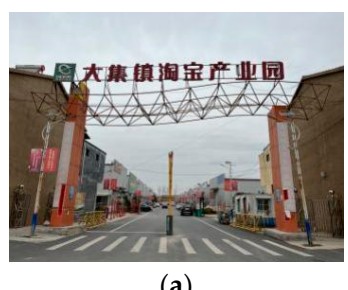
**(a)**

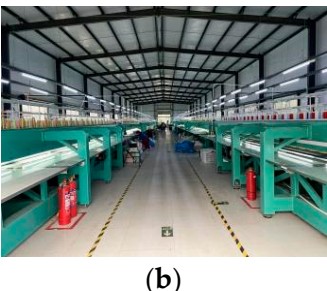
**(b)**

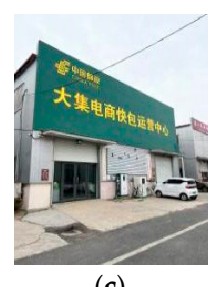
**(c)**

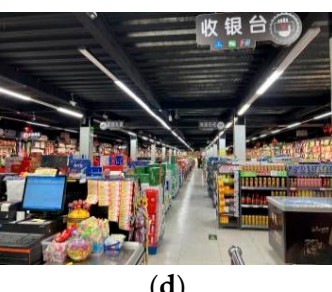
**(d)**

**Figure 7.** Taobao industrial park in Daiji Town (Source: photos taken during the fieldwork). (**a**) Industrial park entrance, (**b**) Standardized plant, (**c**) Express center, (**d**) modern supermarket.

## 5. Conclusions and Final Remarks

The era of digitization has blurred the boundary between urban and rural areas, reconfigured multiple elements and flows among them, and debased cities as the hegemonic centers of capital, talents, wealth, and information. It has enabled trans-border interactions between rural and urban areas and accelerated the urbanization process through intermingling four parts: the promotion of rural non-agricultural industries, the urbanization of farmers' identities and rural life, the innovation of rural governance structures, and spatializing rural areas. Thus, the highlight of this paper is to preemptively deliver an analytical framework that embodies the above four aspects of this new geo-economic phenomenon. Taking Daiji Town in China as an example, it analyzes the details and evidence of digitalization-enabled rural in-situ urbanization. The paper therefore extends the growing body of literature on rural in situ urbanization by synthesizing all interlinked elements instead of simplistic and homogenous accounts focused solely on rural industrialization. Through this integrative research, the main conclusive remarks and the main conclusions are as follows:

Firstly, digitalization connects the rural areas to the production and consumption networks in wider areas, promoting the transformation and prosperity of the rural economy. Wang et al. argue that the impact of e-commerce platforms such as Taobao on rural economic development varies greatly with the relative location of villages to major urban centers [60]. Our case study shows that digitalization can directly connect the scattered rural areas of China to modern society through the internet and logistics network, thus relieving them of the status of being isolated from cities caused by the limitations of local industry, transportation, or other factors and attached to the global urban pipeline network. Digital e-commerce has opened the network market, enabling the original market to extend from rural areas to the outside world and establishing a new network form of production and consumption. The internet platform realizes the rapid integration of online and offline industries, which contributes to the formation of large-scale non-agricultural industrial clusters in rural areas. And the large-scale development of industries further leads to the improvement of relevant service systems, which precisely lay the industrial foundation for urbanization. However, most of the "Taobao villages" developed from the bottom up are mainly engaged in traditional processing industries, with family workshops constituting the main production and operation units. This small-scale operation mode, which is mainly focused on low-end product processing, is not only spatially inefficient but also obtains limited value in the whole value chain.

Secondly, it should be highly noted that the reverse migration of young generations to rural areas becomes the mainstay dynamic of urbanization in the vast rural areas. The new employment opportunities created by e-commerce have greatly enhanced the attractiveness of rural areas to young people. As stated in our research, the reverse immigrants include not only the migrant workers who moved to big cities from the region earlier but also

university graduates and even doctors. These well-educated immigrant entrepreneurs engage their knowledge and skills in the development of rural economies and actively participate in the governance of rural affairs. This could be conceptualized as a new mode of urbanization in China, which should be further advanced considering the population decline and aging, as well as the outmigration of educated people to rural areas.

Thirdly, digitization re-spatializes the vast rural areas. The rural space in various forms anchors digitalized e-commerce, whereby the development of e-commerce has reshaped the rural landscape at the same time. Thus, the land of farmers' own homesteads, as low-cost production space, is of significant assistance for the primitive accumulation of capital for e-commerce practitioners. When the spatial patterns of traditional rural areas are difficult to meet the needs of newly emerging e-commerce industrialization, the modernization and comprehensive expansion of rural space begin to lead to a significant process of spatial urbanization. By probing much deeper into the complex and yet intertwined spatiality of digitalized e-commerce, this emerging body of geographical research will gain new momentum in the context of worldwide disruptions and geopolitical rivalries.

Finally, the digitalization broaches new tracks to wipe off the urban-rural binary in the Global South, far from being epitomized by China's specificity. Nor is it to simplify the urbanization process by turning rural areas into cities through digitalized industrial growth; the goal of this paper is not to elevate digitization as a "one-size-fits-all" panacea to address rural plights. Instead, we hope to enrich and nuance rural studies beyond the extant scholarship to unpack the modernization of rural society by disrupting the urban-rural binary through the lens of digitalized urbanization. The conceptual schema developed in this paper is far from exhaustive; it is urgently needed that cross-fertilizing comparative case research among over 2000 Taobao towns in China be advanced. Further studies with variegated methodological reflexes and cross-disciplinary introspection are needed to enrich the spatiality and temporality of rural digitalized urbanization. As our case study shows, the relationship between urban and rural areas has been reconstructed due to the changing patterns of production and consumption through digitalized e-commerce. When the demand for employment and public services can be met locally in rural areas, it might be the only choice for rural outmigrants to migrate into metropolitan areas or cities. It merits being noted that the self-induced rural construction movement of Chinese farmers against the background of digitalization is a bottom-up urbanization process. However, this bottom-up urbanization has also faced some challenges. Due to the insufficient space capacity of rural areas for this possibility, the phenomenon of empty rural housing waste and the shortage of e-commerce production space coexist in Taobao villages, which need effective governance responses from the innovation of rural land use systems and the compilation and implementation of local rural planning.

**Author Contributions:** Conceptualization, L.L. and T.S.; Methodology, L.L.; Software, L.L.; Writing—original draft, L.L.; Writing—review & editing, T.S. All authors have read and agreed to the published version of the manuscript.

**Funding:** This study was funded by National Social Science Foundation of China [No.19CJY015] and National Natural Science Foundation of China [No.42171180].

**Data Availability Statement:** The data are sourced from fieldwork by authors.

**Conflicts of Interest:** The authors declare no conflict of interest.

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
