# Peer review of "Enabling In-Situ Urbanization through Digitalization"

_land, doi:10.3390/land12091738_

Round 1

Reviewer 1 Report

Introduction: Concepts that you mention are not well defined and quoted. "Planetary urbanization" e.g. was first used as a term by Neil Brenner and Christian Schmid. Also "migration urbanization" is not quoted and defined correctly. (especially lines 65-69)

The research problem should be more clearly stated and be problem-oriented. This research interest is very dsecripitve.

Chapter two: A conceptual framework is promised here in the headline. What we can read is a listing of different thematical aspects. Be more consice when speaking of concepts and theories. Take care to clearly mark why you use which concept and apply it lateron in your analysis.

Methods: you mention that you have conducted "ethnographic fieldwork" (line 217) Yet, you only write of interviews and not of participatory observation or other methods typical for ethnographic fieldwork. 

Figure 2 b) and c) are missing scale bars.

Results: It is very unclear how you quote interviews. You should mark them with quotation marks, so one can understand where exactly they begin.

Overall the study is very descriptive and not critical. You should consider having a more problem-oriented approach. You frame the development of the Taobao villages as only positive. Rethink of what negative effects could be there as well and try to make these voices be heared in your research. What about social and environmental costs? Are taxes paid by the business people also benefitting the whole village? Does this development leave agricultural activities even more unattractive?

- language should be corrected, ideally by a mother tongue speaker or a professional.

Especially plural/singular, definite and indefinite articles and punctuation seems to be an issue. Furthermore, some terms are used in a wrong or complicated way.

Check the language in Figure 1 ("Urbanization in rural")

Author Response

Reviewer 1’s comments are constructive in improving our manuscript. We gratefully appreciate and have made a revision. Following the order of comments, the responses are stated as follows:

Comment 1:

Introduction: Concepts that you mention are not well defined and quoted. "Planetary urbanization" e.g. was first used as a term by Neil Brenner and Christian Schmid. Also "migration urbanization" is not quoted and defined correctly. (especially lines 65-69)

The authors’ answer: Thank you very much for your comments and insightful advice.1) We carefully reviewed the relevant literature of "Planetary urbanization" theory again and added the relevant references of Neil Brenner and Christian Schmid in line 63, such as reference 23. Meanwhile, we checked all the references in the whole paper.2)We have detailed lines 65-69 as following: “How does migration contribute to the urbanization process have attracted more attentions among a vast of theoretical urbanization inquiries [26-28]. Migrants have traditionally been viewed as responsible for excessive urban growth and for urban surplus labour [29-34].”

Comment 2:

The research problem should be more clearly stated and be problem-oriented. This research interest is very dsecripitve.

The authors’ answer: Thanks a lot for your insightful suggestions. These opinions help to improve academic rigor of our article. We have detailed our aims niching to research problem as following: “ This paper has twofold objectives: 1) explore the process and mechanism of digitalization enabling rural urbanization represented by e-commerce, and try to verify them through microscopic empirical research. 2) to nuance the specific case evidence of Daiji Town where digitalization enabled urbanization is starting recently.”

Comment 3:

Chapter two: A conceptual framework is promised here in the headline. What we can read is a listing of different thematical aspects. Be more consice when speaking of concepts and theories. Take care to clearly mark why you use which concept and apply it lateron in your analysis.

The authors’ answer: Thanks for pointing this out. We have carefully revised the conceptual framework. We also tried to avoid many elusive concepts in this part.

Comment 4:

Methods: you mention that you have conducted "ethnographic fieldwork" (line 217) Yet, you only write of interviews and not of participatory observation or other methods typical for ethnographic fieldwork.

The authors’ answer: Thanks for pointing this out. As for the method used by our Group, we emphasized the interview. In fact, the first author lived in the Daiji Town for about 2 months and worked in an embroidery studio mentioned in the paper for 3 weeks, to observe and actually participate in their specific work and life.

Comment 5:

Figure 2 b) and c) are missing scale bars.

The authors’ answer: Thanks a lot for your insightful suggestions. We have added scale bars to Figure 2 b) and c).

Comment 6:

Results: It is very unclear how you quote interviews. You should mark them with quotation marks, so one can understand where exactly they begin.

The authors’ answer: Thanks a lot for pointing this out. We have added quotation marks to all the contents of the interview so that readers can clearly distinguish the contents of the interview.

Comment 7:

Overall the study is very descriptive and not critical. You should consider having a more problem-oriented approach. You frame the development of the Taobao villages as only positive. Rethink of what negative effects could be there as well and try to make these voices be heared in your research. What about social and environmental costs? Are taxes paid by the business people also benefitting the whole village? Does this development leave agricultural activities even more unattractive?

The authors’ answer:Thanks a lot for your insightful suggestions, which enabled us to have more in-depth and critical thinking on the development of Taobao Village. We have supplemented the corresponding parts of the paper and the content is as follows: 1)In addition, informality has also led to a series of safety and environmental issues in rural areas, challenging the government’s governance capability. 2)However, most of the "Taobao villages" developed from the bottom up are mainly engaged in traditional processing industries, with family workshops constituting the main production and operation units. This small-scale operation mode, which is mainly focused on low-end product processing, is not only inefficient, but also fundamentally determines that the "Taobao villages" only obtain the bottom value of the whole industrial chain. 3)In addition, due to poor tax supervision of rural e-commerce, many merchants are evading taxes...However, the associations can do little about tax evasion, nor can the government.

As for agricultural activities, in our interviews, many farmers said that they would still adhere to cultivating their land while operating e-commerce shop, mainly due to their strong feelings towards the land, although they no longer rely on the income from the land.

Comment 8:

Comments on the Quality of English Language

- language should be corrected, ideally by a mother tongue speaker or a professional. Especially plural/singular, definite and indefinite articles and punctuation seems to be an issue. Furthermore, some terms are used in a wrong or complicated way. Check the language in Figure 1 ("Urbanization in rural")

The authors’ answer: Thanks a lot for your insightful suggestions. The English language has been serious proofread. We have checked language in Figure 1 and changed it into “rural in situ urbanization”.Finally, Thank you again for taking the time to review our manuscript.

Reviewer 2 Report

line63: Main - main (no capital is needed)

note the comma before 'and' in lists. 

it would be worthwhile to briefly describe the typical economic composition of Taobao villages and how it has changed over time (the evolution of its non-e-commerce elements, with a brief overview)

would it also be worth elaborating a little more on the distance from which the products have to be sourced?

easy to read and understand. Please note the comments above.

Author Response

Reviewer 2’s comments are constructive in improving our manuscript. We gratefully appreciate and have made a revision. Following the order of comments, the responses are stated as follows:

Comment 1:

line63: Main - main (no capital is needed)

The authors’ answer: Thanks a lot for pointing this out. The English language has been serious proofread.

Comment 2:

note the comma before 'and' in lists.

The authors’ answer: Thanks a lot for pointing this out. The English language has been serious proofread and the comma before "and" has been removed

Comment 3:

it would be worthwhile to briefly describe the typical economic composition of Taobao villages and how it has changed over time (the evolution of its non-e-commerce elements, with a brief overview)

The authors’ answer: Thanks a lot for your insightful suggestions. We have detailed our Profile of Daiji Town as following: “...with aquaculture, planting and processing of agricultural and sideline products as the leading industry in the 1980s. In the late 1990s, Caoxian County actively opened a number of textile factories and established vocational and technical schools to train textile skills and promote economic development, so that villagers mastered the skills of making performance costumes. In 2009, China's "online shopping" craze was in a period of rapid development and the increase of cultural performance activities created a huge segment of the market for performance costumes. Daiji Town became one of the earliest Taobao Towns in China and all the 32 administrative villages under its jurisdiction have been identified as the “Taobao Village”.”

Comment 4:

would it also be worth elaborating a little more on the distance from which the products have to be sourced?

The authors’ answer: Thanks a lot for your insightful suggestions. We have detailed the distance from which the products have to be sourced as following: “The raw materials of these enterprises mainly come from Zhejiang Province, a major textile production area in China.”

Comment 5:

Comments on the Quality of English Language

easy to read and understand. Please note the comments above.

The authors’ answer: Thanks a lot for your suggestions. The English language has been serious proofread according to the comments above.

Comment 6:

Comments and Suggestions for Authors

The topic of the study is interesting and up-to-date.

The authors’ answer: Finally, Thank you again for taking the time to review our manuscript.

Reviewer 3 Report

The topic of the study is interesting and up-to-date.

Title

The title of the manuscript is wrong. Firstly, it does not present a research problem. Secondly, it does not encourage the reader. It is not scientific.

Abstract

The abstract does not have the correct structure that is commonly accepted in scientific journals. A serious mistake is the lack of the purpose of the research and the lack of presentation of how the research methodology was applied.

Introduction

It is generally accepted that introductions fulfill a specific role. Unfortunately, in this study, the introduction does not play this role - there is no explanation why the research was undertaken, what is its purpose. Besides, this introduction is more of a reportage than a scientific study.

Section 2

In this part, a literature review was performed, but it is not known what the purpose of this was - what was wanted to check/prove (no research questions were asked), what conclusions this review led to, etc.

Methodology

The research methodology was not really presented. What research methods and techniques were used and why were they chosen?

Sections 4 and 5

These sections are just descriptions - nothing new/revealing about it. Again, reportage was used rather than research.

Conclusions

On what basis were the conclusions made? Can such subjective statements be multiplied in general?

Summary

Maybe the authors have a good idea for a study, but the manuscript is poorly prepared, has many flaws that do not allow for a positive opinion.

The English language requires serious proofreading.

Author Response

Reviewer 3’s comments are constructive in improving our manuscript. We gratefully appreciate and have made a revision. Following the order of comments, the responses are stated as follows:

Comment 1:

Title

The title of the manuscript is wrong. Firstly, it does not present a research problem. Secondly, it does not encourage the reader. It is not scientific.

The authors’ answer: Thanks a lot for your suggestions. We have revised the paper’s title into“Enabling the in Situ urbanization through digitalization: Evidence from a Taobao village in Shandong Province, China”

Comment 2:

Abstract

The abstract does not have the correct structure that is commonly accepted in scientific journals. A serious mistake is the lack of the purpose of the research and the lack of presentation of how the research methodology was applied.

The authors’ answer: Thank you for these insightful and instructive suggestions. We have revised the abstract section a little bit and highlight the conceptual and methodological novelty.

Comment 3:

Introduction

It is generally accepted that introductions fulfill a specific role. Unfortunately, in this study, the introduction does not play this role - there is no explanation why the research was undertaken, what is its purpose. Besides, this introduction is more of a reportage than a scientific study.

The authors’ answer: Thank you for these insightful and instructive suggestions. We’ve made the change in the introduction part to address your concerns and hope that it is now clearer.

Comment 4:

Section 2

In this part, a literature review was performed, but it is not known what the purpose of this was - what was wanted to check/prove (no research questions were asked), what conclusions this review led to, etc.

The authors’ answer: Thank you for these insightful and instructive suggestions. We revised the literature section by concluding that “Drawing on this conceptual assemblage, this approach delivers the opportunity to transcend the unitary dichotomy between rural/ urban …. through integrating all potent elements into a holistic framework of digitalization enabled urbanization. In tracing the subfield of this dynamic through the following case evidence, the paper places an emphasis on how various elements constitute and sustain the in-situ urbanization driven by digitalization.”

Comment 5:

Methodology

The research methodology was not really presented. What research methods and techniques were used and why were they chosen?

The authors’ answer: Thank you for these suggestions upon the methodology.We revised the methodological section by describing that “The authors of the manuscript conducted ethnographic fieldwork in Cao County and Daiji Village mainly betweenin 2022 and 2023 with some short follow-up trips in 2023. Semi-structured and open-ended interviews with more than 50 government offi-cials, village cadres, enterprise managers, ordinary residents, factory workers and vil-lages were completed accordingly. Snowball sampling was conducted in these ways: first, through local connections by the first author, who lived in the Daiji Town for about 2 months and worked in an embroidery studio for 3 weeks, and second, via offi-cial connections the other coauthor built up through the role of consultant project conducted in Shandong Province. The Table 1 lists the key respondents’ information in this study…….”

Comment 6:

Sections 4 and 5  These sections are just descriptions - nothing new/revealing about it. Again, reportage was used rather than research.

The authors’ answer: Thanks a lot for your insightful suggestions, which enabled us to have more in-depth and critical thinking on the development of Taobao Village. We have supplemented the Sections 4 and 5 with the content as follows: 1)In addition, informality has also led to a series of safety and environmental issues in rural areas, challenging the government’s governance capability. 2)However, most of the "Taobao villages" developed from the bottom up are mainly engaged in traditional processing industries, with family workshops constituting the main production and operation units. This small-scale operation mode, which is mainly focused on low-end product processing, is not only inefficient, but also fundamentally determines that the "Taobao villages" only obtain the bottom value of the whole industrial chain.

Comment 7:

Conclusions

On what basis were the conclusions made? Can such subjective statements be multiplied in general?

The authors’ answer: Thank you for these insightful and instructive suggestions. As for the conclusive part, we have revised it accordingly.

Comment 8:

Summary

Maybe the authors have a good idea for a study, but the manuscript is poorly prepared, has many flaws that do not allow for a positive opinion.

The authors’ answer: We would like to thank the referee again for taking the time to review our manuscript.

Comment 9:

Comments on the Quality of English Language

The English language requires serious proofreading.

The authors’ answer: Thanks a lot for pointing this out. The English language has been serious proofread.

Hopefully, we did not misunderstand the reviewer’s comment. We hope the above responses are adequate and the revisions are acceptable. Thanks a lot for the reviewer’s valuable comments and kind efforts.

the authors